# Decoding Oncofusions: Unveiling Mechanisms, Clinical Impact, and Prospects for Personalized Cancer Therapies

**DOI:** 10.3390/cancers15143678

**Published:** 2023-07-19

**Authors:** Kari Salokas, Giovanna Dashi, Markku Varjosalo

**Affiliations:** Institute of Biotechnology, HiLIFE, University of Helsinki, 00790 Helsinki, Finland; kari.salokas@helsinki.fi (K.S.); giovanna.dashi@helsinki.fi (G.D.)

**Keywords:** cancer, cancer driver, drug discovery, drug target, fusion mutation, immunotherapy, oncofusion, protein kinase, proteomics, sequencing, transcription factor

## Abstract

**Simple Summary:**

Oncofusions, or cancer-associated fusion mutations, are driving forces in cancer development. Advanced sequencing technologies have revolutionized their identification, opening new avenues in cancer research. Oncofusions manipulate cellular signaling pathways and show promise as targets for therapy and diagnostic markers. Further research is needed to understand their functional impact and harness their potential for precision cancer treatment.

**Abstract:**

Cancer-associated gene fusions, also known as oncofusions, have emerged as influential drivers of oncogenesis across a diverse range of cancer types. These genetic events occur via chromosomal translocations, deletions, and inversions, leading to the fusion of previously separate genes. Due to the drastic nature of these mutations, they often result in profound alterations of cellular behavior. The identification of oncofusions has revolutionized cancer research, with advancements in sequencing technologies facilitating the discovery of novel fusion events at an accelerated pace. Oncofusions exert their effects through the manipulation of critical cellular signaling pathways that regulate processes such as proliferation, differentiation, and survival. Extensive investigations have been conducted to understand the roles of oncofusions in solid tumors, leukemias, and lymphomas. Large-scale initiatives, including the Cancer Genome Atlas, have played a pivotal role in unraveling the landscape of oncofusions by characterizing a vast number of cancer samples across different tumor types. While validating the functional relevance of oncofusions remains a challenge, even non-driver mutations can hold significance in cancer treatment. Oncofusions have demonstrated potential value in the context of immunotherapy through the production of neoantigens. Their clinical importance has been observed in both treatment and diagnostic settings, with specific fusion events serving as therapeutic targets or diagnostic markers. However, despite the progress made, there is still considerable untapped potential within the field of oncofusions. Further research and validation efforts are necessary to understand their effects on a functional basis and to exploit the new targeted treatment avenues offered by oncofusions. Through further functional and clinical studies, oncofusions will enable the advancement of precision medicine and the drive towards more effective and specific treatments for cancer patients.

## 1. Introduction

Cancer-associated gene fusions, also known as oncofusions (OFs), are hybrid genes formed when two previously independent genes become juxtaposed (Figure 1). The formation of these fusions can stem from structural rearrangements, transcription read-through of neighboring genes, or the *trans*- and *cis*-splicing of pre-mRNAs. Gene fusions with oncogenic properties are referred to as oncofusions, and often act as driver mutations in different cancers [1]. The identification of the first oncofusion signifies a landmark in the study of cancer [2]. Since then, developments in the applicable methods have enabled an accelerating pace of discovery of new oncofusions in cancer samples, with the pace skyrocketing after massively parallel sequencing (MPS) became viable. Oncofusions have been identified extensively in solid tumors, leukemias, and lymphomas, and in both singular studies [3] and systemic sequencing studies [4,5]. In this review, we seek to describe the current state of the field of oncofusions, with a focus on those oncofusions that manifest as protein-level changes [6].

Oncofusions drive cancer development primarily through two mechanisms: deregulation and the creation of hybrid genes. Deregulation occurs when one gene becomes linked to another’s regulatory region, typically resulting in the overexpression of an otherwise normal gene. The first conclusive evidence for the deregulation mechanism was provided by analyses of Burkitt lymphoma (BL) and chronic myeloid leukemia (CML). These oncofusions link the coding region of the *MYC* oncogene to the regulatory region of immunoglobulin, leading to *MYC* overexpression and subsequently oncogenesis [7,8,9]. The second oncogenic mechanism involves the fusion of the coding regions of two different genes to generate a hybrid gene, resulting in a functional chimeric protein such as *BCR::ABL* [2]. This oncofusion results in a fully functional protein with aberrant kinase activity, and is the best-described model example of abnormal protein function resulting from an oncofusion [10,11,12,13,14,15]. Indeed, oncofusions are known to disturb multiple critical cellular signaling pathways regulating proliferation, differentiation, and survival (Figure 2).

While oncofusions are typically somatic mutations, a few hereditary exceptions have been described [16]. Guided genomic methods such as quantitative real-time PCR (qPCR) and fluorescent in situ hybridization (FISH), together with extensive genomic research efforts like the Cancer Genome Atlas Project (TCGA) and the more recent Japanese Cancer Genome Atlas (JCGA), have contributed significantly to the identification of tens of thousands of oncofusions since the 1960s and 1970s [3,4,5]. The TCGA project has been particularly influential in shedding light on the scale, prevalence, and impact of oncofusions. The project propelled forward the development of both identification strategies and treatment options by molecularly characterizing over 20,000 cancer samples from 33 different cancer types.

The combined knowledge of all known oncofusions supports the significance of deregulation and hybrid gene formation in fusion oncogenicity, although other mechanisms may also play a role. Moreover, some oncofusions can drive cancer development via the inactivation of cancer suppressor genes, for example by coupling them to regulatory elements of genes that are not usually expressed [17,18]. The major challenge with vast numbers of possible, probable, and confirmed oncofusions is validation and relevance: many could be functionally irrelevant or barely expressed passenger mutations, and most have been described as random mutations arising from chance events [19]. However, as several studies highlight, non-driver mutations can still be very valuable for cancer treatment due to their role in producing neoantigens [20,21].

## 2. Oncofusion Formation

Integration of transcriptomic and genomic data has estimated that approximately two thirds of fusion mutations stem from erroneous repair of DNA DSBs [19]. Erroneous DNA DSB repair can lead to genome instability, inducing mutagenesis and genomic rearrangements such as deletions, inversions, duplications, and translocations. All of these genomic rearrangements can result in the formation of oncofusions [22].

Although the majority of fusions transpire in “gene-rich” areas of the genome [19], intragenic fusions have been assessed to constitute only 38% of fusion events [23]. These involve the joining of one gene’s coding sequence with the coding sequence of another, resulting in an in-frame gene fusion such as *BCR::ABL1* or *TMPRSS::ERG*. A large proportion of oncofusions have intergenic regions from at least one fusion partner (Figure 3A), but can sometimes still produce proteins where the 3′ fusion partner is also in-frame [23,24]. Some intergenic oncofusions are the result of a gene with an otherwise weak promoter fusing to a stronger promoter or enhancer (e.g., *IGH-MYC* in BL) [25]. These events are known as promoter or enhancer swapping [23]. In this review, we do not focus on intergenic fusions or fusions which lead to truncation (reviewed recently in [26]). While some fusion genes have distinct partnering patterns in cancer samples (e.g., *FGFR3::TACC3*), most oncofusion-forming genes do not have strict requirements for their partner genes [27].

For the purpose of a general assessment of fusions and participating genes, we downloaded data from tumorfusions.org. Although tumorfusions.org data only reflect data from TCGA and different analyses can produce different results, these data were deemed sufficiently accurate for an overall representation of oncofusions. We conducted an analysis of data downloaded from tumorfusions.org, focusing on key genes participating in fusions. This dataset was then cross-verified and annotated with information from the COSMIC, ChimerDB, and Uniprot databases. The genes were subsequently categorized into four main groups for an in-depth analysis: kinases, transcription factors, non-kinase or TF tumor suppressors, and other oncogenes. The majority of genes participating in fusions, such as *VMP1* and *TACC3*, do not belong to any of these groups (Figure 3B). Of the annotated genes, in-frame fusions, kinases, and transcription factors are equally common, and thus it is no surprise that much research has focused on them in recent years (Figure 3C).

On the kinase side, the majority of the in-frame fusions originated from combinations of *FGFR3*, *BRAF*, *RET*, *NF1*, and *FGFR2*, with 36, 33, 28, 18, and 18 fusions, respectively, with a variety of partners (*FGFR3* combined in particular with *TACC3*) (Figure 4A). A group that has often been identified as a major contributor to fusions, as well as cancers driven by other mutations, is the RTKs, especially the *FGFR* and *NTRK* subfamilies, as well as *RET* and *ALK*.

While kinases hold significant attention in oncofusion research, transcription factors also play a considerable role. For instance, the most common genes involved in oncofusions were ERG, RARA, TFE3, and ETV6, with 65, 28, 18, and 17 fusions, respectively. The group in general was driven by the *TMPRSS2::ERG* fusion, which was overall the most common fusion found, with 60 in-frame fusions identified in the tumorfusions.org TCGA dataset.

Genomic rearrangements are present in roughly half of hematopoietic cancers as well as up to 90% of solid tumors [28], many of which harbor oncofusions [29]. The products of these oncofusions play pivotal roles in tumor evolution and progression [30,31].

## 3. Oncofusions’ Role in Cancer Development

Cancer development has historically been viewed as a slow accumulation of mutations which increases genomic instability, ultimately resulting in both driver and passenger mutations [32]. Instability takes place at the nucleotide level, causing small-scale changes such as point mutations and minor deletions, as well as at the chromosomal level, prompting more substantial rearrangements of entire chromosomal segments [33].

Following the first large-scale identification of oncofusions in MPS studies, it was estimated in 2007 that fusions accounted for roughly 20% of cancer morbidity and represented early events in cancer genesis [31]. More recently, during the JCGA project, known oncofusions from a limited panel of 491 were identified as driver mutations in 12.9% of cancer specimens [5]. Identification varies by cancer type. It is noteworthy that the prevalence of fusions is significant in specific cancers—they are identified in over 90% of all lymphomas [34] and more than 30% of soft-tissue tumors [35]. Other research has suggested that fusions drive 16.5% of human cancers and act as solitary drivers in 1% [36]. These estimates are likely to be reasonably accurate; however, our understanding of oncofusions and their significance in human cancer will undoubtedly trend towards greater precision as sequencing depth increases and the scope of MPS efforts expands.

### Prominent Oncofusions in Cancer

Various kinase and transcription factor fusions are frequently emphasized as driver oncofusions [6,37,38,39,40,41,42,43,44]. In the case of transcription factor fusions, prevalent mechanisms of cancer formation revolve around the DNA-binding domain. Kinase oncofusions often lead to the loss of kinase domain regulation and the acquisition of an alternative dimerization mechanism, such as the coiled-coil domain of *TACC3* in the *FGFR3::TACC3* oncofusion [45].

In addition to gene groups, specific oncofusions have been highlighted as crucial early steps in the initiation of tumorigenesis, or as crucial contributors to tumor morbidity [46,47,48,49]. Furthermore, many oncogenes seem to require the occurrence of specific fusion events to unleash their oncogenic potential. In such instances, there is minimal variability in how the fusion occurs, and they are frequently recognized as recurrent oncofusions in large-scale studies. For example, the oncogenic potential of the *IGH::BCL2* oncofusion is derived from the combination of the anti-apoptotic *BCL2* with the highly expressed immunoglobulin locus of *IGH* [50,51]. A similar mechanism is often present in prostate cancer, where the *TMPRSS2::ERG* oncofusion is found in roughly 50% of tumors and *MAN2A1::FER*, *MTOR::TP52BP1*, and *SLC45A2::AMACR* occur at lower frequencies. The common factor uniting these fusions is the association of an oncogene with a more active promoter region [52,53,54,55].

## 4. Fusion Identification

Although MPS has been used to identify the vast majority of currently known oncofusions, most have yet to be functionally analyzed or validated. Consequently, their mechanisms of action and therapeutic potential remain uncertain. While many oncofusions are strong cancer drivers [17,56,57], others are mutations that have been detected even in normal cells [58,59,60]. Indeed, chimeric RNAs have been proposed to serve as mechanisms of phenotypic plasticity, allowing more variable responses to environmental circumstances without increasing the number of genes [61].

### 4.1. Fusion Callers

The advent of massively parallel sequencing in early 2000s revolutionized the detection of cancer-specific mutations. Genomic and transcriptomic data collected through MPS have contributed to publicly available data repositories such as TCGA and JCGA. The success of treatment approaches targeted towards *NTRK*, *ALK*, and *RET* fusions has heightened the demand for rapid and robust fusion detection. RNA-seq data provide high resolution and complexity with less background noise and a broader dynamic range of gene expression [62]. The RNA-seq data available from TCGA have been used to identify and describe gene fusions in many different studies [63,64,65]. Furthermore, the TCGA data have been utilized in the development of multiple different software packages focused on the detection of oncofusions from RNA-seq data. These fusion caller algorithms have undergone rapid development in the past decade, and play crucial parts in the outcomes of sequencing studies [66,67,68].

Oncofusion identification from sequencing data varies based on the specific fusion caller software employed. Numerous fusion caller algorithms have been published over the past decade [66,67]. Generally, these callers search for reads corresponding to multiple genes and subsequently identify reads that span the fusion breakpoint. However, composite fusions, which involve the combination of more than two genome segments, are usually not considered [24,69]. To best filter actionable and medically relevant oncofusions, various statistical measures are applied alongside a threshold requirement for a minimum number of reads spanning the breakpoint and surrounding regions. Following this, fusion detection proceeds either by assembling sequences into transcripts from which fusions are identified, or by first mapping reads to genes and only then assembling them into transcripts. The former approach excels at recovering fusion transcripts, while the latter has a higher sensitivity [66]. Given the abundance of fusion caller algorithms, variance in their results has prompted the development of metacallers, which often take input from multiple fusion callers and use the top-performing methods to rescore candidate fusions [67,70,71]. Ultimately, metacallers enhance the overall identification confidence by utilizing the most effective algorithms [70]. Through sample collection, sequencing, and bioinformatics analysis, it becomes feasible to examine the oncofusion expression landscape of a certain tumor sample. Identifying oncofusions can inform clinicians about additional targets for supplementary treatment options [72,73].

The significance of analyzing the mutation and fusion landscape is exemplified in breast cancer research, where clinical trials have explored the use of FGFR inhibitors (erdafitinib, AZD4547, PD173074, and infigratinib) in breast cancers that harbor *FGFR* amplifications, mutations, or gene fusions [74,75]. Combining novel oncofusion discovery and the quantification of known oncofusions with rapid testing through sequencing represents a key advantage of employing RNA-seq for clinical diagnosis, treatment, and research in the case of fusion-positive cancers.

### 4.2. Protein-Level Analysis of Oncofusions

At the transcript and protein level, however, fusions become more complex. Even fusion events which DNA-level analysis indicate should produce out-of-frame fusion genes can produce in-frame transcripts and, presumably, proteins [76,77]. Decoding the protein production of oncofusions is an important consideration from both academic and clinical points of view. For instance, detectable and known proteins are required for use as early neoantigens [78] and conducting protein-level functional studies is essential to understanding the behavior of oncofusions. To address the former need, approaches have been developed to work around and complement multiple fusion callers [78].

Numerous earlier studies have pinpointed transcription factors and protein kinases as major gene groups of interest partaking in oncofusions [6,17,31], and these groups make up a large portion of recurring fusions (Figure 4B) as well. A classic example of transcription factor fusion is the *EWSR1::FLI1* fusion, which is a common driver in Ewing sarcoma [79,80]. The most notable protein kinases involved in fusions include the MAST kinases in breast cancer [81], present in 3 to 5% of cases, as well as *RET* in various cancers (particularly lung and thyroid) [76,82,83,84] and other RTKs in a variety of tumors [36,36,85,86,87,88,89,90,91,92,93,94,95,96,97]. Such rearrangements are occasionally cancer-specific, or may be observable in distinctly different cancer types. The fusion of *MAN2A1* and the tyrosine kinase *FER*, for example, activates EGFR-driven proliferation of the cell. Their fusion product *MAN2A1::FER* has been found in several different types of tumors and even in patient serum samples [98,99,100]. The presence of genes participating in fusions in multiple types of tumors probably points towards a competitive advantage conferred by the recurring fusion mutations, and many such recurring oncogenic fusions have been shown to drive cancer evolution (reviewed in more detail by Glenfield and Innan in 2021 [101]).

While in the early decades of oncofusion identification, transcription factors and tyrosine kinases were the most prominent participating gene groups identified, the cast of genes and the variety of their roles is now as wide as the genome: from R-spondins to nucleoporins [102,103,104]. Out of over 20,000 known fusion mutations, the majority are considered passengers rather than driver mutations [19]. The vast majority of fusions identified in MPS studies are non-recurring, and, as such, have been identified as stochastic events [19]. While large-scale MPS efforts have been enormously productive in identifying fusion break points, novel oncofusions and their acting mechanisms are still being identified from MPS data at an accelerating pace [85,86,87,88,105,106,107,108,109,110,111,112,113,114,115,116,117,118,119,120], and while multiple fusion databases are available [64,121,122], a centralized and standardized hub akin to, for example, IntAct [123] for interaction data or UniProt [124] for protein information has yet to emerge.

## 5. Functional Validation of Gene Fusions

Despite the exponential increase in the number of identified gene fusions, only a small number have been extensively characterized functionally. Within the Catalogue of Somatic Mutations in Cancer (COSMIC), only 10 fusions out of 306 detected fusions have been mentioned in more than 50 papers, among which we can find driver fusions identified in the late 1900s such as *BCR::ABL*, *EML4::ALK*, *CCD6::RET*, *EWSR::FLI1, NPM::ALK*, and *TMRSS2::ERG*. Recent analysis of TCGA RNA-seq data has demonstrated identifiable effects of hundreds of protein kinase or transcription factor fusions, suggesting that even mutations thought of as passengers have biological meaning in the case of fusions [6].

Relatively high-throughput functional characterization of oncofusions can be achieved by integrating different omics approaches to gain insight into the roles these oncofusions play in different cancers. Thus far, the major focus has been on genomic and transcriptomic approaches; even though these approaches can be used to determine the molecular environment of the cells, they do not fully capture the dynamic landscape of the intracellular reality.

Proteomic approaches allow the investigation of a highly dynamic and structured intracellular protein space. Extensive genomic perturbations such as gene fusions cause major changes to two different proteins and thus result in an aberrant phenotype leading to cancer. To best understand the actual molecular function of gene fusions, we need to add proteomics to the transcriptomic and genomic information.

Advancements in mass-spectrometry-based methods have enabled the identification and quantification of the majority of the proteome. These methods can be used to investigate post-translational modifications, protein structure, and the composition stoichiometry and topology of protein complexes [125]. Of major interest for chimeric proteins is the application of proteomic approaches to identify fusion-specific protein signaling networks [126,127]. Traditional methodologies, such as Western blotting or co-immunoprecipitation, have been and are still being used to identify fusion-specific signaling pathways. However, the higher throughput of mass-spectrometry-based methods has relegated these methodologies mostly to the role of orthogonal validation of identified interactions.

Oncofusion proteins are considered important for cancer initiation [101], progression, and therapeutic resistance [128,129,130]. It has already been shown in the example of hematological malignancies that the eradication of oncofusions can lead to successful treatment outcomes [31]. The goal of precision medicine for cancer is therapeutic intervention within the processes specific to cancer cells, and this is where the value of in-depth functional investigation of oncofusions lies [131,132]. Understanding the mechanisms of oncofusions can lead to the development of fusion-specific drugs. This is an attractive venture because oncofusions are present only in cancer cells; the drugs could be less toxic and could lead to remarkable treatment results in cases in which oncofusions are the sole cancer drivers [131].

## 6. Oncofusion-Specific Molecular Mechanisms

While deregulation is usually understood to refer to the deregulation of gene expression, in oncofusions there may be as many levels of deregulation as there are regulation. In addition to promoters and enhancers [133], deregulation mechanisms that function via transcript or protein stability have been identified and confirmed in many recurring fusions [134,135,136]. Intergenic fusions often result in dysregulation of the affected genes (referred to as promoter or enhancer swapping or hijacking), whereas intragenic fusions more often result in the dysregulation of protein domains, such as constitutive activation of a kinase domain. The retention of an intact kinase domain and its subsequent activation or deregulation is an important facet of oncofusions, as it contributes directly to cancer development [137].

Some oncofusion proteins have also been proven to be more stable than their wild-type counterparts. For instance, the half-life of the ERG protein is 2.5 h, while that of TMPRSS2::ERG is longer than 5 h [138]. The difference can be as drastic as almost triple in time, as is the case for the PML protein, which has a lifetime of 8 h while that of its fusion PML:RARA is almost 24 h [139]. There are two main mechanisms contributing to the high stability of oncofusions: inactivation of ubiquitin-proteosome degradation and loss of the protein-degradation domain altogether [140,141]. Evading degradation and maintenance of stability makes oncofusions prominent oncogenes.

Many receptor tyrosine kinase (RTK) fusions result in a product that lacks the transmembrane domain and is thus no longer tethered on the cytoplasmic side of the plasma membrane, or that gains a nuclear localization signal from the fusion partner. As an example, ALK fusions in lung cancer have long been observed to have multiple different phenotypes depending on the fusion partner, including cytoplasmic, nuclear, and phase-separated [142,143,144,145,146].

RTK fusions are indeed often good examples of oncofusion-induced kinase activation. A prime example is the identification of NTRK oncofusions as key drivers across diverse cancer types. The NTRK gene family encodes three TRK receptors: TRKa, TRKb, and TRKc. Each of the receptors is composed of an extracellular domain, a transmembrane region, and an intracellular domain (ICD) which contains the tyrosine kinase domain and its activation loop. Within the ICD, five critical tyrosine residues regulate the receptor’s kinase activity: three on the activation loop and two flanking the kinase domain. While the activation loop sites govern kinase activity, the remaining two tyrosine residues act as phosphorylation-dependent docking sites for various adapters and enzymes essential to the receptor’s signaling functions.

ETV6::NTRK3 fusions include the HLH domain from ETV6, which in WT-ETV6 acts as a dimerization domain and can therefore serve as a way for the kinase domain of NTRK3 to dimerize in the absence of the usual activation stimuli of the receptor. Indeed, fusions are the most common mechanism of oncogenic NTRK activation, and these fusions are found in ~0.31% of adult tumors (and 0.34% of pediatric ones) [147]. However, in certain cancer types, they are nearly always present [147]. Their presence in cancers such as colorectal cancer (CRC) is a well-defined diagnostic marker and indicates probable resistance to EGFR-inhibitor treatment [91,148]. Similarly, RET fusions occur at a very low rate in CRC, but are diagnostic due to being mutually exclusive with other driver mutations [149].

In addition to the diagnostic role of kinases involved in oncofusions, their impact can generate measurable changes that lead to more rapid diagnosis. For example, PLAG1-fusion-driven cancers feature a significant increase in Wnt and IGF signaling [150]. Similarly, the *BCR::ABL1* fusion in CML functions via constitutive kinase activation, which in turn drives proliferation by enhancing downstream signaling pathways [151,152,153]. The pathways affected are distinct from the similar *SFPQ::ABL1* fusion [154]; thus, clinicians can acquire valuable information regarding potential causes and treatment options for a tumor prior to employing RNA or DNA sequencing methods.

## 7. The Importance of Oncofusion-Specific Molecular Mechanisms

Over the past two decades, intensive research has focused on uncovering the functional roles of a select group of oncogenic fusions. More recently, large-scale sequencing efforts such as TCGA and JCGA have detected hundreds of gene fusions in different cancers with unknown oncogenic mechanisms. Oncofusions promote oncogenesis at the transcriptional and phenotypic levels through a multitude of different mechanisms [155]. A robust understanding of the roles of OFs in different cancers calls for the integration of high-throughput omics methods. Although large-scale genomic fusion identification has been an ongoing effort, with whole repositories full of the resulting data, functional studies are sorely needed. The following examples emphasize the intricate nature of oncogenesis driven by gene fusions.

### 7.1. BCR::ABL

The discovery of the Philadelphia chromosome 1 (Ph1) and *BCR::ABL* gene fusion sparked substantial interest in uncovering and exploring the roles of gene fusions in cancer. Despite the limited resources and methodologies of the time, the discovery of Ph1 revolutionized the diagnosis and treatment of many cancers [2]. Decades after the discovery of Ph1, advanced cytogenic and cloning methods revealed that the chromosomal translocation between chromosome 9 and 22 resulting in the formation of Ph1 forms a chimeric fusion of the breakpoint cluster region (*BCR*) and Abelson tyrosine-protein kinase (*ABL*) genes [156,157]. Simultaneously, Naldini et al. [158] were trying to identify the functional effect of Ph1 and the resulting chimeric protein. With the use of Ph1+ leukemic cell lines and simple proteomic methods such as Western blotting, immunoprecipitation, and radioactive kinase activity assays, the researchers determined that there are two isoforms of chimeric *BCR::ABL*, p210 and p190. P120 is a hallmark of chronic myeloid leukemia and p190 occurs in Ph1+ acute lymphoblastic leukemia [159]. They also showed that the ABL found in these chimeras is constitutively activated and that it phosphorylates smaller substrates [158].

In contrast to the wild-type ABL found in both the nucleus and the cytoplasm, the BCR::ABL fusion protein is strictly cytoplasmic, where it undergoes its primary functional interactions [160]. Physiological ABL in the nucleus inhibits cell growth, a function that is abolished once it is fused with BCR. However, inactivated BCR::ABL can enter the nucleus and, once entrapped there, it results in the apoptosis of all fusion-positive cells [161,162]. The kinase domain’s overactivation and the phosphorylation of BCR aberrantly activates the MAPK, JAK/STAT, and PI3K/AKT pathways, enabling the cell to evade apoptosis while promoting proliferation [163]. The discovery of *BCR::ABL* guided subsequent genomic discoveries which resulted in the identification of hundreds of fusions in the following years.

### 7.2. ETV6::NTRK3

Among the oncofusions of the TRK family, *ETV6::NTRK3* (E6N3) demonstrates transformative capabilities in multiple cell types and can be found in numerous hematological and solid tumors [164,165,166]. *E6N3* is the product of a chromosomal rearrangement t(12;15) (p13;q25) that has been identified in congenital fibrosarcoma, secretory breast carcinoma, acute myeloblastic leukaemia, and gastrointestinal stromal tumors. *E6N3* encodes a chimeric oncoprotein consisting of the N-terminal SAM dimerization domain from ETV6 and the C-terminal protein tyrosine kinase (PTK) domain from NTRK3. The constitutively active kinase domain in E6N3 has strong transforming abilities in vitro and in vivo.

*ETV6* and *NTRK3* exhibit preferred breakpoint sites—*ETV6* up to exon 4 or exon 5, *NTRK3* from either exon 12 or exon 13—resulting in four different *E6N3* fusion variants. The most frequently detected is the canonical *E6N3* variant, which features a breakpoint between ETV6 exon 5 and NTRK3 exon 13 [167]. The different fusion variants are distinguished by the presence or absence of ETV6 exon 5 and NTRK exon 12. The ETV6 exon 5 forms the interdomain between SAM and ETS necessary for transcriptional repression, while the NTRK3 exon 12 contains the tyrosine 516 (Y516), an interaction site necessary for downstream signaling.

The most common variant, known as canonical E6N3, promotes downstream Ras/ERK and PI3K/AKT. The active kinase domain of NTRK3 combined with the intact SAM oligomerization domain of ETV6 induces ligand-independent PTK dimerization followed by kinase activation, leading to the stimulation of diverse signaling pathways [168].

In the early 2000s, Tognon et al. studied the *E6N3* oncofusion extensively, using different virally transduced cell models for Northern and Western blotting experiments, co-immunoprecipitation, and different mutagenesis assays to describe how E6N3 exerts its oncogenic activity. They found that both Mek1 and Akt are phosphorylated in a serum- independent manner in E6N3-expressing cells, leading to constitutive activation of the Ras–Erk1/2 and PI3K–Akt pathways. The subsequent overactivation of these pathways can increase the expression of cyclin D1, promoting cell growth through the evasion of cell cycle arrest [169].

In their effort to describe the exact mechanism underlying the functionality of E6N3, researchers discovered that E6N3 does not interact with previously known NTRK3-associating proteins such as Shc, Grb2, PI3K p85, ABL, SH2Bβ, Src, or Ship2, but rather requires a functional insulin-like growth factor 1 (IGF1) axis to induce oncogenesis [170]. Subsequent in vivo and in vitro functional assays revealed that E6N3 exerts its function through a different pathway by interacting with IGF1R/INSR signaling. A decade later, Tognon et al. demonstrated that inhibition of the IGFR/INSR pathway in E6N3 in vivo and in vitro models led to reduced tumor growth and cyclin D1 expression, suggesting a reliable alternative target to TRK inhibitors or for combinatorial drug treatment [171].

### 7.3. NPM::ALK

Over 80% of ALK-positive anaplastic large-cell lymphomas (ALK + ALCL) are characterized by a chromosomal translocation t(2;5)(p23;q35) resulting in the nucleophosmin-anaplastic lymphoma kinase NPM-ALK (NA) fusion protein. The presence of an intact NPM domain causes the dimerization of the ALK kinase and leads to constitutive activation of the kinase domain—a crucial event for oncogenesis. As NPM is a nuclear protein necessary for chain elongation during DNA replication, it potentially mediates the nuclear localization of NA, thereby exposing the fusion protein to a new assortment of kinase substrates.

Known ALK interactions have been tested in NA+ cell lines to identify the pathways NA uses to exert its influence. While NA interacts with characteristic ALK partners such as Shc and IRS, these interactions have been deemed non-essential for the transformation potential of the oncofusion [172,173]. Early affinity purification and chromatography separation methods revealed that NA interacts with the SH domains of Grb2 and PLC-gamma, with PLC-gamma exhibiting the strongest interaction. Mutagenesis experiments further demonstrated that disrupting the interaction with PLC-gamma via mutation of the tyrosine 664 of NA was sufficient to inhibit mitogenesis [172].

Subsequent studies probing the function of NA indicated that different pathways are utilized for the anti-apoptotic properties of NA-positive ALCL. High-throughput proteomic methods revealed that NA associates with PI-3-kinase and Src kinase pp60 (c-Src) to evade apoptosis and promote proliferation. Additionally, it was discovered that NA activates JAK2 and phosphorylates STAT3 in the JAK–STAT pathway [174]. Mutagenesis studies inhibiting NA’s interactions with PI3K, Src, and JAK2 resulted in diminished cell growth in NA-positive samples, which highlighted these targets for the treatment of NA+ cancers [172,175,176].

### 7.4. TMPRSS2::ERG

In human prostate cancer, TMPRSS2::ERG (T2E) is the most commonly observed oncofusion. T2E arises from an intrachromosomal deletion which results in the 3′ end (C-terminal) of Transmembrane serine protease 2 (*TMPRSS2*) merging with the 5′ end (N-terminal) of ETS-related gene (*ERG*). *ERG* is a proto-oncogene and part of the ETS transcription factor family, which has been found to be consistently overexpressed in ~50% of patients with prostate cancer (PC). TMPRSS2 is an androgen-regulated protein kinase essential in proteolytic cascades for the normal physiological function of the prostate [177]. The T2E fusion serves as a central malignant regulatory switch in PCa by inhibiting androgen-receptor-driven differentiation, which leads to the transformation of epithelial prostate cells into undifferentiated, embryonic stem-like cells [55].

T2E induces ERG overexpression due to the androgen-responsive promoter of the *TMPRSS2* gene. Overexpressed ERG activates epithelial-to-mesenchymal transition (EMT) pathways, which leads to cancer progression and metastasis. ERG achieves this through different pathways: it promotes metastasis by upregulating the matrix metalloproteinases and using the ZEB1/ZEB2 axis to promote mesenchymal cell transition. Additionally, ERG hyperactivates inflammatory pathways by interacting with TLR-4, which subsequently activates NF-kb, increasing the transcription of target genes which trigger tumor growth and progression [177].

Furthermore, ERG activates the *EZH2* promoter, leading to the inhibition of tumor suppressor genes such as *NKX3.1* and resulting in constitutive *T2E* expression [55,178]. Patient RNA-seq analysis has revealed that T2E-induced overexpression of ERG directly correlates with different Wnt signals as well as the upregulation of frizzled receptors such as FZD4. Wnt ligands bind to frizzled receptors and trigger a signaling cascade which stabilizes B-catenin. Stabilized B-catenin acts as a transcription factor to increase gene expression related to EMT [179,180].

Prostate cancer prognosis and mechanisms are not solely dependent on *ERG* expression. T2E is frequently followed by PTEN and TP53 loss, leading to overactivation of the PI3K/Akt signaling pathways and progression into invasive carcinoma [181,182]. Recent studies demonstrated that ERG increases expression of IGF1R via activation of the transcription factor Sp1. Further mass spectrometry experiments subsequently revealed a possible new multimeric complex in a PC cell line involved in ERG-mediated effects on the IGF1R signaling axis [183]. Since targeting androgen receptors alone often results in resistance to treatment, identifying different interactors and pathways affected is a key method in the development of novel treatment strategies. In conclusion, T2E and ERG overexpression promote cell proliferation, survival, and angiogenesis while modulating tumor progression and aggressiveness. Clinical treatment of PC has already taken advantage of the information uncovered thus far, indicating that a richer treatment landscape will become available for clinicians as the amount of information grows [184].

The increased workload required to describe the functionality of OFs using low-throughput methods, together with the high occurrence of specific fusions such as *BCR::ABL* or *TMPRSS2::ERG*, explains why functional studies in the past have focused on a handful of prominent fusions. Simultaneously, advanced genomic and transcriptomic approaches have been successfully used to understand the function of OFs at the genetic and transcriptomic levels. However, these methods do not reflect the direct intracellular consequences or detailed molecular mechanisms of OFs. To depict a complete picture of OF function, the inclusion of functional proteomic studies alongside high-throughput genomics and transcriptomics is imperative. The aforementioned examples illustrate how the integration of different methods leads to a deeper understanding and characterization of the molecular properties of gene fusions. This yields clinically relevant insights, leading to better diagnosis and new avenues for the development of treatment.

## 8. Gene Fusions as Neoantigens

The genetic alterations that accumulate in cancerous cells give rise to tumor-specific antigens (TSA), or neoantigens. Neoantigens are presented by MHC (major histocompatibility complex) proteins to T cells, subsequently eliciting an immune response. Innate immune cells kill cancer cells; however, cancer cells inevitably acquire methods to evade the immune response. A multitude of therapeutic strategies are under development to boost the anti-cancer response of the immune system, including tumor vaccines, transference of tumor-infiltrating lymphocytes, TCR-transduced T cells, immune checkpoint blockades (ICBs), bispecific antibodies, CAR-T cells, and modulators of the tumor microenvironment [185,186].

Neoantigens originate from the mutational burden of the cancer cells. They can be shared between different cancers [187] or cancer-specific [188]. They hold promise for precision oncology because they are cancer-specific and absent in healthy cells. Neoantigens arise at the genomic level from SNVs (single-nucleotide variants), INDELs (base insertions and deletions), and gene fusions, whereas at the transcriptomic level, they can arise from alternative splicing, polyadenylation, RNA editing, and non-coding regions. Additionally, protein-level alterations such as the dysregulation of proteins or post-translational modifications can also produce neoantigens [186]. Even though transcriptomic and proteomic determinants may offer high specificity and a large pool of targets, they remain mostly unexplored. Currently, INDELS and gene fusions represent the best-understood sources of neoantigens [186]. Similarly to neoantigens, fusions also offer other fusion-specific opportunities for treatment, such as abnormal splice sites and other sequence features that may represent therapeutic vulnerabilities in cancer cells [189].

### 8.1. Neoantigen Immunogenicity

Among these sources of neoantigens, gene fusions hold significant advantages due to their elevated immunogenicity: They offer more neoantigens per mutation, they rarely occur in healthy cells, and they can be shared between tumors. A comprehensive analysis of TCGA data demonstrated that gene fusions generate 6-fold more neoantigens and 11-fold more fusion-specific neoantigens compared to SNVs and INDELs [190,191]. Gene fusions are a rich resource for highly immunogenic neoantigens, as illustrated by only 5.8% of the shared recurring fusion neoantigens detected in the TCGA having a low immunogenic potential [186,190]. Another possible result of an intergenic fusion event is the creation of an out-of-frame fusion protein, in which the 3′ gene experiences a frame shift and contributes nonsense to the resulting protein sequence. Previously, such out-of-frame fusions have been shown to be stable, i.e., not rapidly degraded, but the specific function of the out-of-frame fragment remains unknown [192]. Fusions from this group are often assumed to be inactivating mutations [193], or are labeled passenger mutations or ignored altogether in analysis [135,194]. However, stable out-of-frame fusions with drastically distorted downstream amino acid sequences and passenger mutations possess the highest immunogenicity [190].

Kinase and oncogene fusions are likely to retain their ORFs to maintain their oncogenic function and promote cell growth, compared to tumor suppressors and passenger mutations [190]. Oncofusions containing oncogenes have a very low immunogenicity, which indicates that they undergo selection pressure during tumorigenesis. Although studies thus far have often focused on driver mutations, it is interesting that cell growth and immune resistance increase the amount of passenger fusions, which makes them highly immunogenic and thus very interesting for vaccine development.

In 32.2% of TCGA patients, highly immunogenic neoantigens were generated due to oncofusions, rendering them suitable for neoantigen vaccine development [190]. Pediatric patients with metastatic Ewing sarcoma and alveolar rhabdomyosarcoma generally have a 5-year survival rate of less than 25%. However, an already concluded clinical trial using EWS-FLI1 and PAX3/FKHR breakpoint peptides (NCT00001566) as a vaccine in combination with supplementary immunotherapies resulted in a 43% 5-year overall survival rate [195]. Since then, there have been follow-up clinical trials using similar peptide vaccines (NCT00001564). A BCR::ABL fusion peptide vaccine has also been tested in clinical trials (NCT00267085). Currently, one clinical trial testing a peptide vaccine against DNAJB1::PRKACA for patients with fibrolamellar hepatocellular carcinoma (NCT04248569) is in the recruitment phase. The low number of clinical trials investigating fusion vaccines as single treatments is likely attributable to the aggressiveness of fusion-positive cancers [191].

### 8.2. Treatment Potential for T-Cell Therapies

The potential of gene fusions in adoptive T-cell therapy has been predominantly investigated in vitro. Highly immunogenic neoantigens produced by *TMPRSS2::ERG2* in pancreatic cancer, *DEK::AFF2* in metastatic head and neck cancer, and *CBFB::MYH11* in acute myeloid leukemia have been used to generate neoantigen-reactive T-cells. Cytotoxic T cells specific for fusion antigens, tested on cell models, in vivo, and in xenografts, have exhibited promising results for the use of fusion neoantigens in development of neoantigen reactive (NRT), T-cell receptor (TCR), and chimeric antigen receptor (CAR) T cells [196,197,198]. In short, fusions, particularly ones previously dismissed as of little interest, hold huge potential for cancer immunotherapy. A challenge remains in identifying broadly applicable neoantigens with high immunogenic potential, rather than patient-specific ones.

## 9. Clinical Implications

Oncofusions have long been tempting targets for cancer therapeutics and diagnostics due to their significance and cancer-specificity [199]. The world of cancer diagnostics and treatment is always looking for new biomarkers that can help to diagnose accurately or early, indicate prognostic or therapeutic attributes, or indicate a relapse or response, among other information [200]. For example, the *BCR::ABL1* rearrangement of the Philadelphia chromosome has become an early biomarker for chronic and acute myeloid leukemias, as well as for monitoring in the case of a relapse or incomplete response [201]; in prostate cancer, *TMPRSS2::ERG* fusion detection has been suggested as a screening tool [202]. However, the detection of fusions beyond diagnostic sampling has also been shown to yield results, and fusions contribute to the mutational burden of tumors post-treatment [203]. Moreover, oncofusions are some of the most cancer-specific biomarkers identified thus far, rendering them immensely valuable tools to detect both tumors and their driver genes. Although many fusions have also been detected in normal tissues [58], even passenger fusion mutations can still provide powerful and cancer-specific neoantigens for therapeutic use [204]. Overall, the development of methods relating to fusion detection and targeting is changing the treatment landscape of multiple cancer types [27,143,205,206,207].

As a diagnostic tool, DNA sequencing alone has been shown to be unreliable for determining prognostic outcomes and treatment efficacy in at least some fusion-driven cancers [76,77,208]. However, for in-frame fusions, DNA-seq works roughly as well as RNA-seq, and is still an important tool particularly for patients who do not have sufficient tumor sample mass for concurrent DNA- and RNA-seq [76]. For example, in lung cancer, one of the main activation mechanisms of the common driver gene *BRAF* is a fusion mutation [209,210]. *BRAF* fusions have also been identified as mechanisms of secondary resistance after treatment of EGFR-driven cancer with tyrosine kinase inhibitors (TKIs) [211]. Fusions as mechanisms of treatment resistance independent of original driver genes have been reported for multiple types of cancer [94,211,212,213,214], and some secondary RTK oncofusions in particular act as mechanisms of acquired TKI resistance [94,212,215].

Specific fusions are associated with particularly difficult cancers. For example, while *NRG1* (which encodes the ligand for ErbB2 and -3 RTKs) fusions are found in 0.2% of solid tumors [216], they can be found in 25% of cases of the uncommon invasive mucinous lung adenocarcinoma [217]. The presence of the *TMPRSS2::ERG* fusion in prostate cancer is associated in part with cancer recurrence, and commonly corresponds to a more aggressive malignancy and overall poorer clinical prognosis [31,54,218,219,220,221,222]. Generic attributes such as fusion type (intra- or intergenic) alone have thus far not been shown to be reliable characteristics for prediction, e.g., of treatment outcomes or cancer progression [223].

However, detection in a clinical setting mostly relies on low-throughput methods such as FISH or immunohistochemistry, which require relatively high amounts of tissue sample and can only test for one fusion at a time [224,225]. The exception to this is qPCR, which, while still low-throughput, is more sensitive and can be somewhat multiplexed [224,225]. More recently, RNA-seq methods have been shown to be both rapid and equally sensitive for fusion detection, while being able to identify any fusion genes present in the patient sample with little starting material and with impressive specificity [96,226,227]. The methods applied can be successfully deployed for all oncofusions; however, those with low expression levels remain a challenge [226,228]. This concern can be somewhat mitigated by using targeted MPS methods [227]. In general, RNA-seq approaches still benefit from faster turnaround time, the ability to detect multiple kinds of mutations from a single sample, and requiring a small amount of sample material [224]. In addition, it may be possible to gain additional insights about the tumor microenvironment as well [229].

The presence and specific detection of oncofusions may guide the treatment process from diagnosis to prognosis, and may help to inform clinical actions as well as set expectations for patient response and further treatment [224]. Given the cancer-specific nature of oncofusions, targeting them and their protein products has been proposed as a strategy to minimize off-target side effects, which are very often present in cancer therapeutics [95]. Oncofusion-specific therapies have indeed already been developed; however, all of them target RTKs, partly due to the strong oncogenic potential of RTKs and partly due to the availability of existing kinase inhibitors such as imatinib [89,230,231,232,233,234]. Unfortunately, inhibitor treatments are also effective at generating selective pressure, which can lead to evolution of the cancer into a version that is resistant to the applied therapy, thereby buying valuable time, but ultimately not curing the cancer [151,235,236,237]. Recently, a lower dosage of inhibitors has been suggested as a therapeutic regimen aimed at containing the tumor rather than eliminating it and causing higher selective pressure [238]. Meanwhile, later-generation tyrosine kinase inhibitors have been shown to somewhat mitigate the risk of acquired resistance [151]. Oncofusion-specific treatment options, therefore, are still in their early days as a way of curing the disease, while the treatment of kinase fusions with kinase inhibitors is an evolving, mature field with many drugs already available [42,44,239,240,241].

## 10. Conclusions and Future Directions

Integrative omics studies hold immense potential for the development of cancer-specific drugs. An example of a targeted treatment is the breakthrough discovery of imatinib. Imatinib is an ABL kinase inhibitor targeting BCR::ABL and it was the first example of a targeted cancer therapy. This discovery prompted the exponential development of kinase inhibitors (KI) for many mutated kinases, including OFs. The development of omics approaches and their application in clinical settings will expand the growth of precision medicine, especially in the sphere of tumors harboring gene fusions. A practical example can be seen in a recent study combing high-throughput patient-derived organoid drug testing and sequencing conducted by Murumägi et al. in 2021 [242]. RNA sequencing revealed the presence of a STERN-ALK fusion in a malignant peritoneal mesothelioma (MpeM) in a pediatric patient. Patient cells were used to generate organoid cultures to test over 500 drugs. The results identified ALK inhibitors as efficient drugs with variable efficacy between different inhibitor types. Interestingly, most cytotoxic drugs had low efficacy, but multiple other effective drugs that inhibited ALK downstream pathways were identified. The standard first-line treatment for mPeM is cytoreductive surgery (CRS) with heated intraperitoneal chemotherapy (HIPEC), and the prognosis is less than 12 months after diagnosis. However, after detailed molecular profiling, the patient was treated with the ALK inhibitor citrozininb in combination with chemotherapy; at the time of writing, they had reached complete cancer remission with no recurrence 3 years after diagnosis [242].

Although kinase inhibition is an effective and currently the only more or less oncofusion-specific approach, it inevitably culminates in drug resistance [89,236]. A potentially fruitful alternative avenue for small-molecule inhibitors are cancer-specific protein–protein interactions, for example on the IL-6/JAK/STAT axis. FDA-approved agents for inflammatory diseases targeting the IL-6/JAK/STAT axis are being evaluated together with novel STAT3-specific inhibitors for the treatment of hematological malignancies and solid tumors [243]. Several trials are also exploring multi-axis targeting in cancer therapeutics. For instance, a 2022 phase 1 clinical trial (NCT05010005) investigated a combinatorial treatment of the JAK inhibitor ruxolitinib and the PI3K inhibitor duvelisib for different lymphomas, including ALCL. Another clinical trial study in 2018 tested the combination therapy of ruxolitinib and the BCR::ABL inhibitor nilotinib in CML patients, resulting in positive outcomes which warranted progress to the next phase [244]. The evaluation of targetable signaling axes specific to oncogenes is paramount to steering cancer therapeutics toward precision medicine and fostering the development of novel drugs and drug repurposing.

Some oncofusions are left without an effective treatment option. Cancers harboring transcription factor fusions such as EWS::FLI1 in Ewing sarcoma lack easily targetable pockets in their structure. Furthermore, many kinases in OFs can experience partial or complete kinase domain loss, which can render kinase inhibitors ineffective in treating these cancers. The limitation of inhibitor drugs can be surpassed by the development of directed protein-degradation molecules. In the era of development of molecular glues and proteolysis-targeting chimeras (PROTACs), attacking these undruggable targets is becoming an achievable goal [245,246]. Protein degraders are a promising treatment option not only for the direct eradication of oncoproteins but also for overcoming drug resistance. With the advancements in the field since the first PROTAC, ARV-471, entered clinical trials, many degrader compounds have been developed to target different proteins from kinases to transcription regulators [245,246]. These treatments cause a significant drop in the stability of the mutated proteins [247].

The most up-to-date research on the use of protein degraders for oncofusion proteins is still in the preclinical phase. Nevertheless, the results are promising. PROTACs targeting BCR-ABL [248,249] and ALK fusions [250,251] already show potent degradation activity. Since the PROTAC small-molecule warheads are made of existing kinase inhibitors, they currently focus on kinase fusions. In these instances, PROTACs not only degrade the oncofusions, but also inhibit drug resistance [252,253]. The development of protein degraders such as molecular glues [254] and PROTACs shows promising potential for targeting oncofusion proteins, overcoming drug resistance, and expanding treatment options in cancer research.

Overcoming resistance and achieving precision therapy towards only cancer cells is an important, yet still overall unmet clinical need in the cancer field [212]. Considering the role of fusions as primary drivers and mechanisms of resistance to treatment, more functional information for the development of alternative treatment strategies is required. However, in contrast to troublesome resistance mechanisms, a promising use case for fusions is in their role as diagnostic markers, and neoantigens for precise immunotherapy [204]. For patient monitoring, continuous sampling, especially during treatment progress, may be required to identify additional mutations, including fusions (discussed in more detail by Powers [200]), which could be utilized to target the cancer instead of the initially identified driver mutations.

To conclude, addressing the challenge of resistance and advancing precision therapy for cancer cells requires a deeper understanding of oncofusions. Information about oncofusions at the sequencing level is already spectacular and increasing steadily. However, the protein-level implications of oncofusions and their specific mechanisms of interaction remain largely unknown. OF-specific features such as splice sites and neoantigens are already attractive potential targets for current and future treatment regimens. Therefore, the ever-increasing amount of patient-specific data together with future functional studies relating to the protein level of OFs seems promising for ushering in a wealth of therapeutic avenues for future exploitation and, ultimately, an increasing proportion of cancers that are successfully treated.

## Figures and Tables

**Figure 1 cancers-15-03678-f001:**
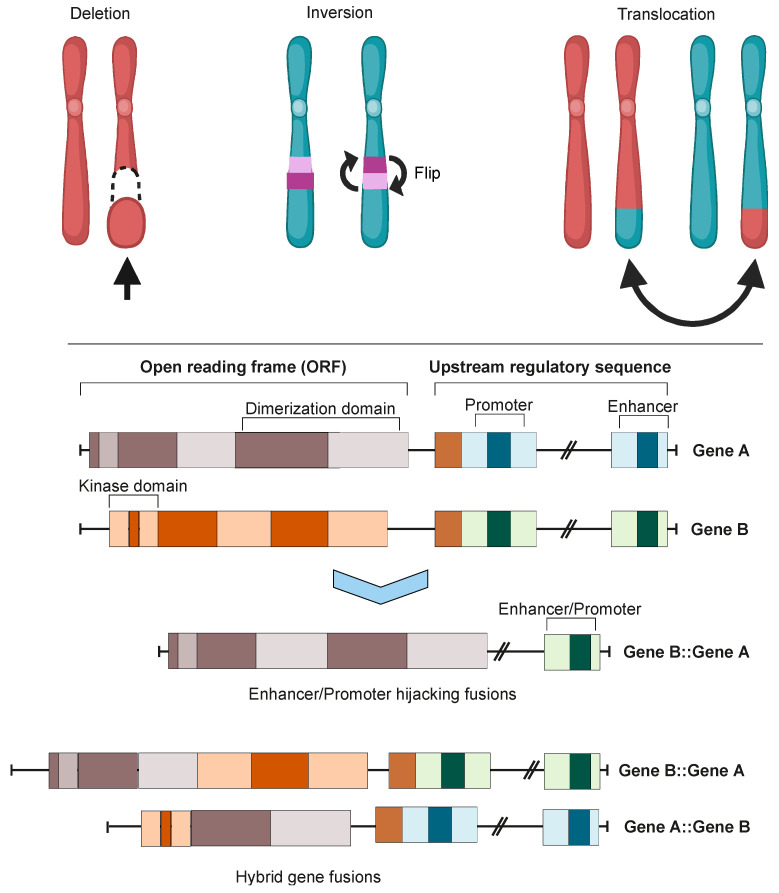
Chromosomal instability produces gene fusions. **Top panel**: chromosomal rearrangements leading to fusion mutations. **Bottom panel**: types of gene fusions. Depending on the specific fusion sites in both genes, the result can be either a hybrid gene fusion or a promoter- or enhancer-hijacking fusion. In both cases, the produced proteins can be either in-frame or out-of-frame. Images were created using BioRender.com.

**Figure 2 cancers-15-03678-f002:**
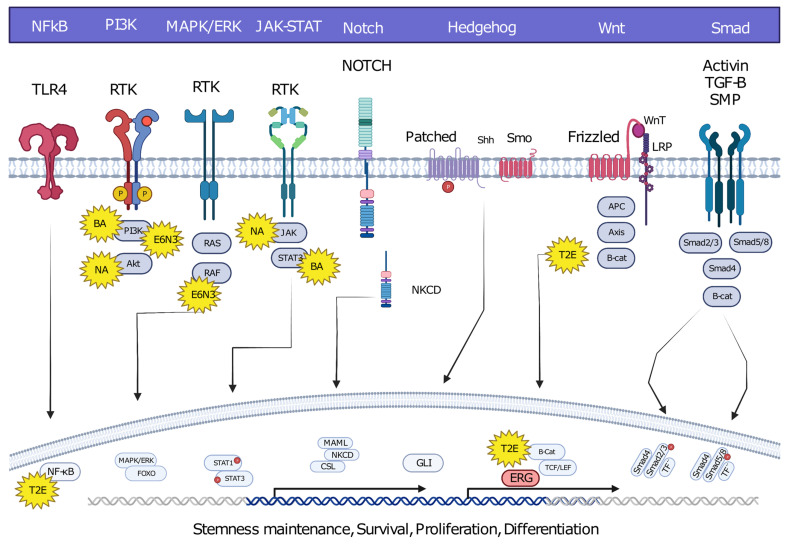
Hybrid oncofusions disrupt many receptor-dependent signaling pathways, whether downstream or upstream. Common fusion-affected pathways are highlighted along with well-known fusions affecting the corresponding pathways. BA: BCR::ABL, E6N3: ETV6::NTRK3, NA: NPM::ALK, T2E: TMPRESS2::ERG. Image was created using BioRender.com.

**Figure 3 cancers-15-03678-f003:**
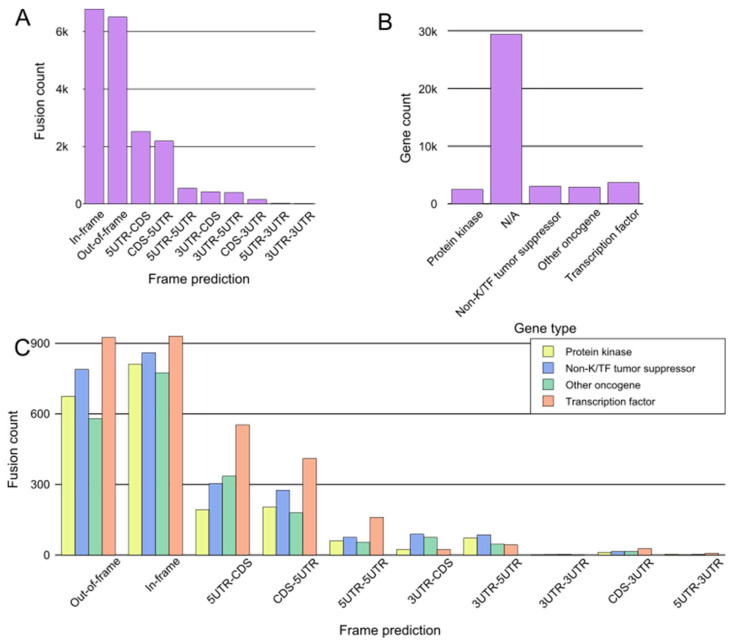
General assessment of fusions from tumorfusions.org database. (**A**) Count of fusions based on frame prediction. (**B**) Fusion distribution based on gene type. N/A denotes genes which could not be categorized definitively as protein kinases, transcription factors, tumor suppressors, or other oncogenes. (**C**) Distribution of fusions based on frame prediction and gene type. Only fusions that could be assigned to one of the four gene groups are included.

**Figure 4 cancers-15-03678-f004:**
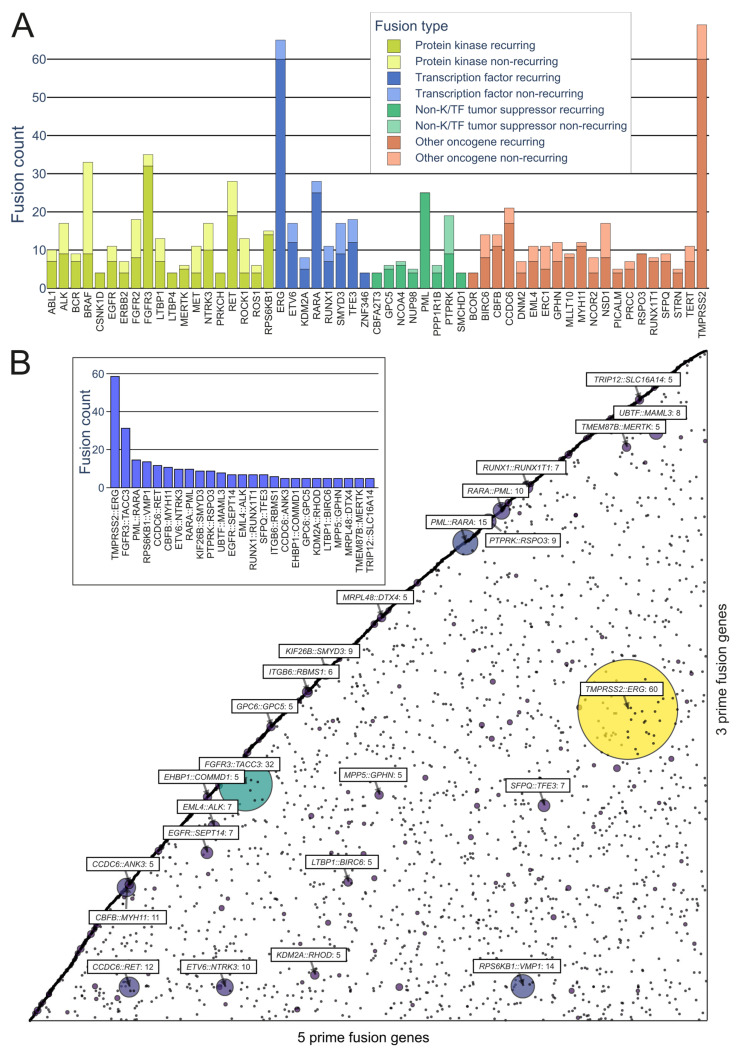
Representation of recurring fusions from tumorfusions.org data. (**A**) Count of recurring and non-recurring fusions per gene in each gene type category. Included are genes present in at least 5 recurring fusions. (**B**) Scatter plot of all in-frame fusions. 5 prime genes are on the *x* axis and 3 prime genes are on the *y* axis. Size and color of the bubbles correspond to the number of fusions in the tumorfusions.org dataset. Gene pairs with 5 or more recurring fusions identified are highlighted. Inset: highlighted fusions in bar plot format.

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
