# Peer review of "Decoding Oncofusions: Unveiling Mechanisms, Clinical Impact, and Prospects for Personalized Cancer Therapies"

_cancers, 2023, doi:10.3390/cancers15143678_

Round 1
Reviewer 1 Report
This paper comprehensively introduces the oncofusions from several aspects, including oncofusion formation, their roles in cancer development, methods for oncofusion identification, approaches for the functional validation, and molecular mechanisms of oncofusions in leading to cancer predisposition, as well as their potential as neo-antigens in cancer diagnosis and cancer therapy. Finally, the paper discussed the clinical implications of oncofusions, and the future directions of research and clinical transformation. The review is well written and provides some conclusions on summarized data and provides insights into the clinical implication of oncofusions.
Some minor corrections will improve the quality of the manuscript:
1. In the bottom panel of Figure 1, Please describe in the figure legend what each color box means. In the Open reading frame region it seems that one color box represents an exon. However, it is hard to imagine what the color box means in the promoter and enhancer region. Too many color boxes make the schematic illustration complicated and hard to understand. I recommend coloring Gene A with only one color and Gene B with another color. The promoter and enhancer region can be marked with different transparency to the open reading frame region.
2. Section 3 focuses on the ‘oncofusion role in cancer development’, however, it only includes one subsection named ‘3.1. Prominent fusion genes’. The subsection title is not in line with the section topic logically. They need to be reconsidered and rearranged.
3. Page 5, line 4, ‘We conducted ananalysis of data downloaded from tumorfusions.org,’ the ‘ananalysis’ seems should be ‘an analysis’.
4. The ‘5p gene’ and ‘3p gene’ in Figure 4 legend need to be explained for their meanings.
5. Change the title of section 7 to ‘Oncofusion specific molecular mechanisms’.
6. In section 4, the text ‘While many oncofusions are strong… … without increasing the number of genes’ matches the section 3 topic better. I recommend the author move this part to section 3.
7. The second paragraph in section 4.2 seems doesn’t fit the section topic ‘Limitations and challenges of MPS-driven identification efforts’. Delete this paragraph or move it somewhere else.
8. Page 13, ‘which results in the 5’ end of Transmembrane serine protease 2 (TMPRSS2) merging with the 3’ end of ETS-related gene (ERG).’ The description leads to the misunderstanding that the oncofusion is formed with ERG at the N terminus. Please revise the description.
9. RNA-seq rather than RNAseq is the common expression. Please revise.
Author Response
We sincerely thank the editor and the reviewers for taking the time to read our manuscript and for providing the valuable feedback. We have addressed all concerns that were raised. We are confident that these amendments have improved our manuscript.
Reviewer 1.
Some minor corrections will improve the quality of the manuscript:
This paper comprehensively introduces the oncofusions from several aspects, including oncofusion formation, their roles in cancer development, methods for oncofusion identification, approaches for the functional validation, and molecular mechanisms of oncofusions in leading to cancer predisposition, as well as their potential as neo-antigens in cancer diagnosis and cancer therapy. Finally, the paper discussed the clinical implications of oncofusions, and the future directions of research and clinical transformation. The review is well written and provides some conclusions on summarized data and provides insights into the clinical implication of oncofusions.
Some minor corrections will improve the quality of the manuscript:
- In the bottom panel of Figure 1, Please describe in the figure legend what each color box means. In the Open reading frame region it seems that one color box represents an exon. However, it is hard to imagine what the color box means in the promoter and enhancer region. Too many color boxes make the schematic illustration complicated and hard to understand. I recommend coloring Gene A with only one color and Gene B with another color. The promoter and enhancer region can be marked with different transparency to the open reading frame region.
We appreciate the reviewer’s comment and we have changed the figure so that there are single colours between gene A and gene B making it easier to follow.
- Section 3 focuses on the ‘oncofusion role in cancer development’, however, it only includes one subsection named ‘3.1. Prominent fusion genes’. The subsection title is not in line with the section topic logically. They need to be reconsidered and rearranged.
We appreciate the reviewer’s comment and taking it into consideration we have changed the subtitles so that they are in line with the section topic.
- Page 5, line 4, ‘We conducted ananalysis of data downloaded from tumorfusions.org,’ the ‘ananalysis’ seems should be ‘an analysis’.
We appreciate the reviewer pointing out a misspelling mistake which was corrected.
- The ‘5p gene’ and ‘3p gene’ in Figure 4 legend need to be explained for their meanings.
We appreciate the reviewer noticing a mistake in the figure, the “p” stands for prime and we have changed the naming in the figure to the “5 prime” and “3 prime”.
- Change the title of section 7 to ‘Oncofusion specific molecular mechanisms’.
We thank the reviewer for the section title advice, we have accepted it and made the change.
- In section 4, the text ‘While many oncofusions are strong… … without increasing the number of genes’ matches the section 3 topic better. I recommend the author move this part to section 3.
We appreciate the reviewer’s comment, however in the context of the fusion identification section, we wanted to emphasise that many of the detected fusions are actually not driver mutations and accurate and statistically stringent methods are required to select the driver fusions that is why we believe this sentence is a better fit for section 4.
- The second paragraph in section 4.2 seems doesn’t fit the section topic ‘Limitations and challenges of MPS-driven identification efforts’. Delete this paragraph or move it somewhere else.
We appreciate the reviewer’s comment, after additional reading we believe that a more suitable title for this subsection might be : “Protein level analysis of oncofusions” instead of ‘Limitations and challenges of MPS-driven identification efforts’.
- Page 13, ‘which results in the 5’ end of Transmembrane serine protease 2 (TMPRSS2) merging with the 3’ end of ETS-related gene (ERG).’ The description leads to the misunderstanding that the oncofusion is formed with ERG at the N terminus. Please revise the description.
We appreciate the reviewer’s comment and we have changed the text as follows: T2E arises from an intrachromosomal deletion, which results in the 3’ end (C-terminal) of Transmembrane serine protease 2 (TMPRSS2) merging with the 5’ end (N-terminal) of ETS-related gene (ERG).
- RNA-seq rather than RNAseq is the common expression. Please revise.
We appreciate the reviewer’s comment and we have changed to RNA-seq throughout the review.
Reviewer 2 Report
Dear authors,
The paper "Decoding Oncofusions: Unveiling Mechanisms, Clinical Impact, and Prospects for Personalized Cancer Therapies" by Kari Salokas, Giovanna Dashi and Markku Varjosalo was revised as requested previously and the manuscript fulfils the high standards required for publication in “Cancers” (ISSN 2072-6694).
We analyzed the originality, scientific quality, relevance to the field, presentation and adequacy of the references of the paper.
This manuscript is acceptable in present form.
Author Response
We thank the reviewer for his/her comments.
Reviewer 3 Report
Comments manuscript: cancers-2505797
Review
Title: Decoding Oncofusions: Unveiling Mechanisms, Clinical Impact, and
Prospects for Personalized Cancer Therapies
It is a manuscript that concentrates important information on oncofusions, a relevant topic for the area of cancer of any type.
The authors work their manuscript mainly around the idea that the gene fusions cause major changes in quantity or quality of the two different involved proteins or regions, and thus result in an aberrant phenotype leading to cancer.
However, it is well known that chromosomal rearrangements cause disturbances at the level of the three-dimensional organization of the genome. Chromosomes and the sequences they carry, work by organizing themselves into chromosomal territories with exquisite precision, such that Oncofusions detected as gross chromosome rearrangement must be handled not only as a major change in the function of the two regions involved -which is valid- but it must also be considered as a major change in the three-dimensional structure of the nuclear organization. This can necessarily generate disturbances beyond the two regions involved... this is not discussed by the authors. It is suggested to include this aspect in the manuscript.
Other observations:
P2. Use oncofusions or OFs throughout the manuscript. If you use OFs, please put the abbreviation from the first time it is used.
P5, Figure 3. In A the Y axis is in numbers, in B is with “k”, it must be homogeneous.
P6, Figure 4. This figure can be confusing if a good explanation of what is being graphed is not shown in the figure caption. In addition, this reviewer does not understand what its usefulness is. It is suggested to remove it or explain it well and use it..
Other details: “5p genes are on the x-axis”, Capital X. In this figure, the X and Y axes are denoted as "5p" and "3p" which should be 5' and 3' or 5 prime and 3 prime. It is suggested to put it in the usual way, since in the cytogenetic nomenclature: 5p= short arms of chromosome 5 and 3p= short arms of chromosome 3.
Author Response
Reviewer 3.
It is a manuscript that concentrates important information on oncofusions, a relevant topic for the area of cancer of any type.
The authors work their manuscript mainly around the idea that the gene fusions cause major changes in quantity or quality of the two different involved proteins or regions, and thus result in an aberrant phenotype leading to cancer.
- However, it is well known that chromosomal rearrangements cause disturbances at the level of the three-dimensional organization of the genome. Chromosomes and the sequences they carry, work by organizing themselves into chromosomal territories with exquisite precision, such that Oncofusions detected as gross chromosome rearrangement must be handled not only as a major change in the function of the two regions involved -which is valid- but it must also be considered as a major change in the three-dimensional structure of the nuclear organization. This can necessarily generate disturbances beyond the two regions involved... this is not discussed by the authors. It is suggested to include this aspect in the manuscript.
We appreciate the reviewer’s thought-out comment and agree that cancer is a very heterogenous disease and that there are many factors influencing oncogenesis in addition to gene fusions. However in the context of this review mentioning the role of chromosomal translocations on nuclear organization and its influence on oncogenesis is a bit outside of the scope and space limits of our article and topic focusing on the role of oncogenic gene fusions specifically.
Other observations:
P2. Use oncofusions or OFs throughout the manuscript. If you use OFs, please put the abbreviation from the first time it is used.
We appreciate the reviewer’s comment and have added the abbreviation at the first mention of oncofusions.
P5, Figure 3. In A the Y axis is in numbers, in B is with “k”, it must be homogeneous.
We appreciate the reviewer’s comment and have changed the figure so that it is homogenous throughout.
P6, Figure 4. This figure can be confusing if a good explanation of what is being graphed is not shown in the figure caption. In addition, this reviewer does not understand what its usefulness is. It is suggested to remove it or explain it well and use it.
We appreciate the reviewer’s comment and want to emphasise how figure 4 is made from real life sequencing data accessible through tumorfusions.org. Here we demonstrate that there are many recurring fusions in cancer which make them interesting targets for pan cancer research. Furthermore, we nicely visualize which fusions are highly recurrent such as TMPRSS2::ERG or FGFR3::TACC3.
Other details: “5p genes are on the x-axis”, Capital X. In this figure, the X and Y axes are denoted as "5p" and "3p" which should be 5' and 3' or 5 prime and 3 prime. It is suggested to put it in the usual way, since in the cytogenetic nomenclature: 5p= short arms of chromosome 5 and 3p= short arms of chromosome 3.
We appreciate the reviewer’s comment we have made the necessary changes to the Figure 4.